# Differentiation of Forest Stands by Susceptibility to Folivores: A Retrospective Analysis of Time Series of Annual Tree Rings with Application of the Fluctuation-Dissipation Theorem

**Vladislav Soukhovolsky** [1,*]**, Polina Krasnoperova** [2]**, Anton Kovalev** [3] **, Irina Sviderskaya** [2]**, Olga Tarasova** [2]**, Yulia Ivanova** [4]**, Yuriy Akhanaev** [5] **and Vyacheslav Martemyanov** [5,*]

1   V.N. Sukachev Institute of Forest, Siberian Branch of Russian Academy of Sciences (SB RAS), Krasnoyarsk 660036, Russia
2   Department of Ecology and Nature Management, Siberian Federal University, Krasnoyarsk 660041, Russia
3   Krasnoyarsk Scientific Center SB RAS, Krasnoyarsk 660036, Russia
4   Institute of Biophysics SB RAS, Krasnoyarsk 660036, Russia
5   Institute of Systematics and Ecology of Animals SB RAS, Novosibirsk 630091, Russia
*   Correspondence: soukhovolsky@yandex.ru (V.S.); martemyanov79@yahoo.com (V.M.)

**Abstract:** This study analyzed the relationship between characteristics of annual tree ring time series and the intensity of attacks on forest stands by forest insects. Using tenets of the fluctuation–dissipation theorem (which is widely used in physics), time series parameters are proposed that can help to assess the susceptibility of a forest stand to insect pests. The proposed approach was applied to evaluate differences in parameters of tree ring widths among outbreaks of the pine looper, Siberian silk moth, and spongy moth. A comparison of trees characteristics between outbreak locations and undamaged forest stands (control) showed that the tested parameters statistically significantly differed between the outbreak locations and control stands and can be used to assess the risk of pest outbreaks in forest stands.

**Keywords:** forest insect; outbreak of mass reproduction; forest stand; external factor; resistance; annual tree ring; spectrum; dendrochronology; *Dendrolimus sibiricus*; *Lymantria dispar*; *Bupalus piniarius*

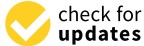



## 1. Introduction

An important problem in research on the spatial structure of outbreak locations of forest insects' population outbreaks is the identification of the reasons for the emergence of an initial outbreak site in a certain small habitat. Does an outbreak start for random reasons, or does the initial outbreak site arise due to characteristics of the starter forest stand? If insects choose the territory of a future outbreak according to some parameters and/or are able to maximally implement their biotic potential, then it can be hypothesized that future outbreak locations and intact forest stands differ in physiological parameters of trees before the outbreak? Nonetheless, because a researcher almost always arrives at an outbreak location only after the outbreak has begun, it is difficult to assess the pre-outbreak state of trees in outbreak locations and outside them. In the analysis of the differences in tree properties between future outbreak locations and intact stands, two retrospective characteristics can be employed. These are (1) distance characteristics, such as the normalized difference vegetation index of forest stands before an outbreak [1], and (2) parameters of a series of annual tree ring widths (TRWs), also in a period before the outbreak. At the same time, however, it is necessary to determine which parameters of the series of TRWs are suitable for analysis when an investigator evaluates the susceptibility of certain forest stands to insect attacks.

Dendrochronology methods are the most famous in the field of research on annual rings of woody plants. For instance, TRW has been used to reconstruct climate variability

over centuries or millennia in various parts of the world [2–4]. In this context, trees that dominate a stand are usually chosen for analysis in order to reduce the effects of interactions among trees [5]. Aside from the climate, insect attacks can strongly influence a series of TRWs of various tree species [6–10]. Changes in TRWs have been documented after the actions of such insects as the eastern spruce budworm (*Choristoneura fumiferana* Clem.) [11,12], the forest tent caterpillar (*Malacosoma disstria* Hubner) [13], the larch sawfly (*Pristiphora erichsonii* Httg.) [7,14], and the grey larch budmoth (*Zeiraphera diniana or griseana* Gn.) [10,15]. Nevertheless, these studies have not analyzed the characteristics of TRW prior to an outbreak, and addressed only changes in TRW during an outbreak, when insects do affect the anatomical features of annual tree rings and of xylem by decreasing biomass accumulation, lumen area, and the number cells, thereby leading to the formation of narrow annual rings for several years after the outbreak's start [16].

Changes in TRWs before the damage can be examined from the standpoint of regulation processes of woody stands' growth. In the course of tree growth, TRWs show an age-related trend, namely, a decrease in TRWs with the age of a tree [5]. Therefore, if the age trend is removed from the data, then we can analyze the regulation of annual ring growth as some steady-state process with the long-term average of zero. During such a process, as usually happens in regulated systems, deviations from the steady state occur under the influence of external factors, and the steady state is restored by negative feedback [17]. In the first approximation, processes of the high-frequency regulation of annual-ring growth can be characterized by two indices: characteristic feedback time and fluctuation amplitude of the first differences in TRWs relative to the average value, which is zero.

In physical systems, to describe a response of a system to an external stimulus, the so-called fluctuation–dissipation theorem (FDT) is used, according to which the spectrum of the system's characteristics before the stimulus is related to dissipative changes in it after the action of the stimulus [18–22].

The relations postulated by the FDT can be expressed in a simplified form as

$$S(f) = KA''(f) \tag{1}$$

where $S(f)$ is the spectral power of a characteristic of the system before external stimulus, $A''(f)$ is the imaginary part of the system response to the stimulus at frequency $f$ (this part characterizes the susceptibility of the system), and $K$ is some system-specific constant.

If this approach is used to describe an interaction of a tree with insects, then we can hypothesize the existence of a relationship between characteristics of the annual-ring spectrum of the tree during a certain period before an outbreak and the severity of damage to the tree by insects after the onset of an outbreak. Our study is aimed at determining whether the FDT principles are suitable for describing the risks of mass attacks by defoliator insects on forest stands depending on spectral characteristics of a time series of TRWs before an outbreak. In the context of this work, hereinafter we will use the term susceptibility of forest stands to defoliators, meaning a response of plants, including a combination of attractive/repellant properties of a feed plant and/or its antifeedant properties. Models based on the FDT are tested here on three systems: "drooping birch (*Betula pendula* Roth.)—spongy moth (*Lymantria dispar* L.)", "Siberian silk moth (*Dendrolimus sibiricus* Tscetv)—fir (*Abies sibirica*) / Siberian pine (*Pinus sibirica*)", and "pine looper (*Bupalus piniarius* L.)—Scots pine (*P. sylvestris*)".

## 2. Materials and Methods

The spongy moth, *Lymantria dispar* L., is widespread in the forests of Europe, Asia Minor, Turkestan, the Caucasus, all of Siberia, Japan, and North America. This species is the main pest of *Betula pendula* Roth. in the forests of Western Siberia. In Siberia, the so-called Asian race *L. dispar* is present; its characteristic feature is the good flight ability of individuals, which allows them to travel long distances and form secondary outbreaks over vast territories [23]. In the conditions of Western Siberia, outbreaks of this species are observed every 8–10 years. In the present work, population outbreaks were studied

in stands of *B. pendula* in the Novosibirsk Region. The age of the trees is 70–80 years, the density about 500 trees ha$^{-1}$, the average height is 26 m, and the average diameter is 33 cm.

Pine looper *Bupalus piniarius* L. is one of the mass pests of *Pinus sylvestris* L., with an extensive range from England and Scotland to Eastern Siberia. *B. piniarius* is the only phyllophage species that has outbreaks in the pine forests of Siberia. In the territory of Eastern Siberia, outbreaks of this species occur at intervals of 12–14 years [24]. We studied outbreaks of mass reproduction of this species in pine forests in the south of Krasnoyarsk Territory. The stand density was about 1000 trees ha$^{-1}$, the age 100–120 years, the average height 20 m, and the average diameter 20 cm.

The Siberian moth *Dendrolimus sibiricus* Tschetv. is a dangerous pest of conifer forests in Siberia and the Far East, damaging stands of *Abies sibitica* Ledeb, *Larix sibirica* Ledeb, and *Pinus sibirica* Du Tour. The high flight ability of individuals leads to population outbreaks over wide areas. Such outbreaks occur in Siberia every 10–12 years [25]. Studies of trees in the *D. sibiricus* outbreak area were conducted in the forests of the southern Krasnoyarsk Region, including *A. sibirica* (age 80–100 years, height 16–18 m, average diameter 32 cm) and *P. sibirica* (age 100–120 years, average height 14 m, average diameter 35 cm). The outbreak developed in a deserted area with no roads and access to the outbreak area was only possible by helicopter. These difficulties of access to the damage zone limited the amount of obtained data.

To reveal a possible relationship between characteristics of processes regulating the growth of annual rings, a retrospective analysis was performed on TRWs in trees from active-outbreak locations (susceptible forest stands) and in trees that were not under a mass insect attack.

The first region to be analyzed included sample plots in the south of Western Siberia (Novosibirsk region, Russia), where in 2021 outbreaks of mass reproduction of *L. dispar* L. were registered, and severe grazing of tree crowns by its caterpillars was observed. In the immediate vicinity (to adjust the data for effects of weather/edaphic conditions), sample plots were set up on which no severe crown grazing was present. Figure 1 shows locations of spongy moth outbreaks and control intact stands in the studied regions. Each location contained heavily damaged (label "D") and slightly damaged or undamaged (label "K") plots.

Sample plots in separated birch stands of the Novosibirsk region were chosen at the beginning of July 2021 via a ground level visual assessment of crown grazing by spongy moth caterpillars, and by judging by the presence or absence of caterpillars on the trunks and in the crowns of the trees. Eight sample plots were selected that had been damaged by spongy moth caterpillars (pp03D–pp10D, Figure 1), as well as eight control plots, where the damage was either absent or small (pp03K–pp10K, Figure 1). To estimate the severity of leaf damage in a whole stand, the forest stands were surveyed by means of an unmanned aerial vehicle. The distance between the control and damaged stands did not exceed 200 m.

The composition of the forest stands in all the examined sample plots was the same: 10B, and the age of trees in the stands was approximately 60–70 years. In the region of spongy moth outbreaks, 10–12 cores were taken with an increment borer in each damaged plot, and 7–8 cores in each control plot.

The second region to be analyzed included sample plots in an area of mass reproduction of the Siberian silk moth *D. sibiricus* in mixed fir–cedar–birch stands in the Sayan Mountains (ip01D and ip012K, Figure 1). According to remote observations, severe damage to fir and Siberian pine by the pest in these forest stands began in 2019. Fir stands on the outbreak-affected and on undamaged plots were similar in taxational characteristics (height, diameter, stand density, and age [~80–100 years]).

The third region included sample plots damaged by the pine looper *B. piniarius* and a control intact stand in the territory of Krasnoturanskiy pine forest (kp01D, kp02D, and kp02K, Figure 1). An outbreak of mass reproduction of the pine looper took place during 1976–1978 [26]. Forest stand observations began in 1978 and continued for nearly 40 years (until 2017) after the end of the outbreak. Trees in the damage and control locations had

an average age of 80–100 years, a mean height of 18 m, and an average diameter at chest height of 20 cm. In the area of mass reproduction, according to assessments from 1979, tree crowns were damaged from 50% to 100%, but by the time our core samples were taken in 2015, only trees with crown damage not exceeding 25%–50% remained in the stand. In the control stand, no damage to the crown was visible.

From trees in all three outbreak regions, cores were collected with an increment borer. Pine cores in the two outbreak locations in Krasnoturanskiy pine forest (31 cores) and in the control intact stand (17 cores) were taken in 2015. Birch core samples from the outbreak-affected plots (78 cores) and in the control (63 cores) were collected in 2021. Core samples from fir and Siberian pine in the region of the outbreak of the Siberian silk moth (15 cores each) and in control intact forest stands (10 cores) were taken in 2021. After drying, the cores were mounted on substrates and polished with sandpaper of gradually diminishing grain size (down to P400 . . . 600 according to ISO-6344). TRW was measured on a LinTab semi-automatic complex (Rinntech, Germany). The compiled annual-ring series were cross-dated using the CDendro software 9.8.1 [27] to find missing rings and reject those whose radial growth course differed significantly from that typical for the stand in question.

In all three regions, for the examined trees in the outbreak locations and in the control, the time series of annual rings were analyzed corresponding to the 2–20 years preceding the onset of a population outbreak. The choice of lengths of the series was determined by the need to obtain statistically significant characteristics of spectra of the series. To remove a time trend from the data, we switched from series of TRWs $L$ to series of first differences $\Delta L$ of TRWs: $\Delta L(i0 = L(i+1) - L(i))$.

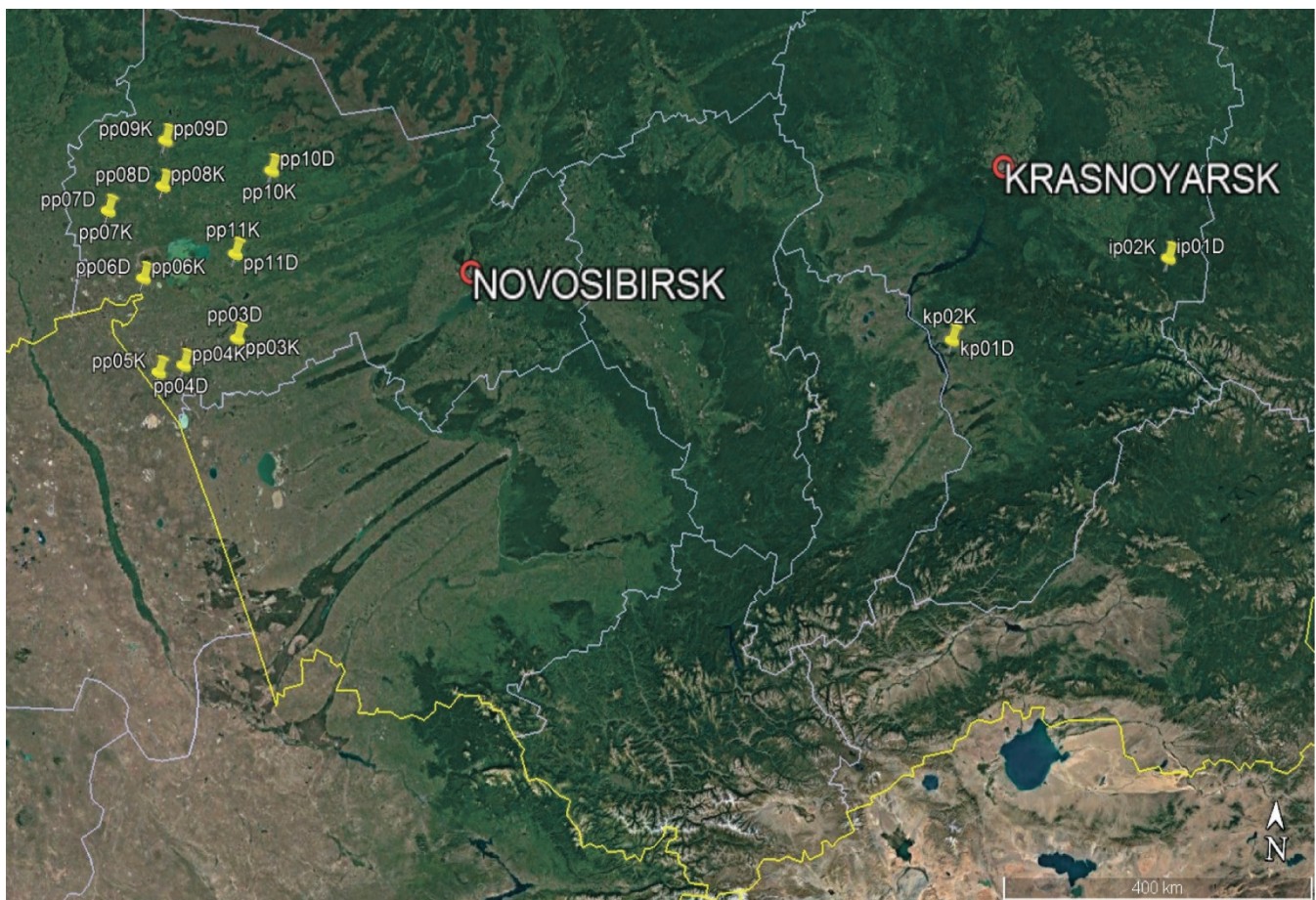

**Figure 1.** Locations of insect attack outbreaks ("D" at the end of a label) and of control intact forest stands ("K" at the end of a label) in the studied regions.

For spectral analysis, the time series used must be stationary [28]. Time series of the annual rings' widths are non-stationary, but the class of non-stationary time series can be reduced to stationary ones using the operation of taking difference [29]. We used the Augmented Dickey–Fuller Test (ADF) [29] to evaluate the stationarity of such series.

The first differences series of the annual rings' widths considered according to the test used can be characterized as stationary, and consequently the spectral analysis of these series is possible. This procedure was performed on all trees on all sample plots in all outbreak regions. In further analyses, only series of the first differences were used.

Because the mean of the first differences series was zero, standard deviation for each series was used to estimate the variance of the first differences in these series. The spectrum of each time series was calculated using the authors' custom-designed software (FFT ver.1.1) written in the Borland Delphi 7 environment using standard fast Fourier transform algorithms [30]. The frequency of maximal spectral density $f_{max}$ was chosen as a characteristic of the spectrum. The software allows the computation of characteristics of spectra of all trees on a sample plot in batch mode.

## 3. Results

Typical shapes of series of TRWs L and of first differences ΔL of TRWs during 2000–2020 for a tree at outbreak location pp03D are presented in Figure 2.

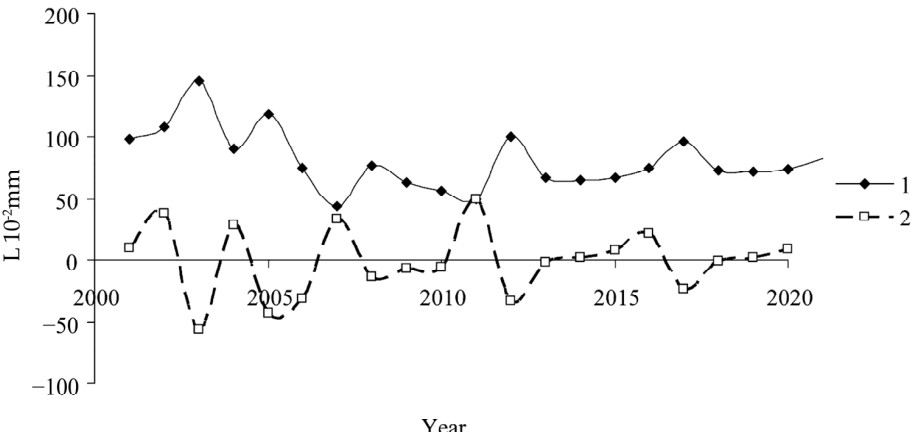

**Figure 2.** Series of TRWs L (1) and first differences ΔL of TRWs (2) of drooping birch for tree "d03" on sample plot pp03D.

For drooping birch, there were no significant differences in average TRWs <L> and in their average standard deviations <s> between any outbreak locations and control forest stands (Figure 3).

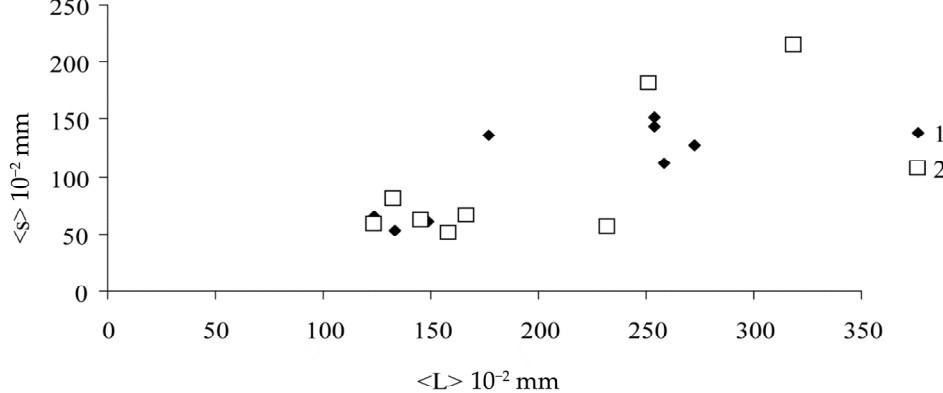

**Figure 3.** Average TRWs <L> and their average standard deviations <s> for drooping birch trees in outbreak locations (1) and control stands (2).

For series of the first differences in TRWs, whose mean long-term value was close to 0, standard deviations were calculated next. Because the series of first differences in TRWs was stationary, one should calculate the spectral density of such a series. Typical spectra of first differences series in TRWs at a spongy moth outbreak location and in a control stand are given in Figure 4.

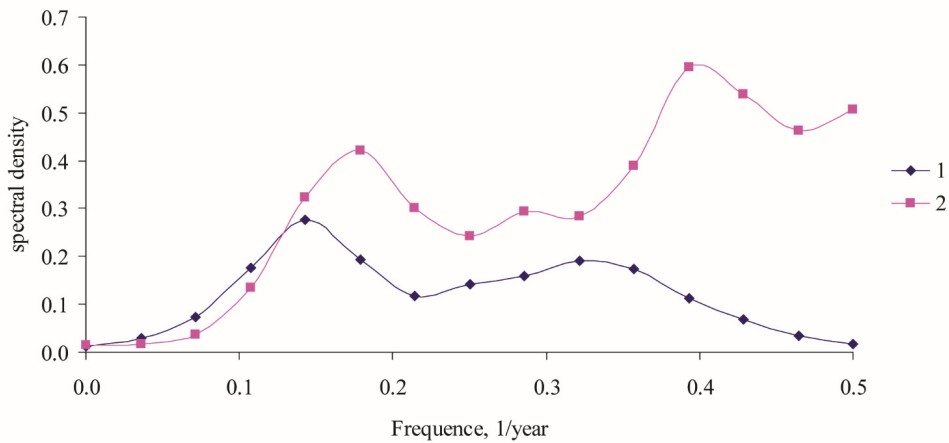

**Figure 4.** Spectra of series of first differences in birch TRWs. 1: tree d01 on sample plot pp03D ($f_{max}$ = 0.16) at a location of an outbreak of gypsy moth mass reproduction; 2: tree k02 on sample plot pp03K ($f_{max}$ = 0.40) in a control undamaged stand.

It should be noted that the comparison of the first difference series of annual rings widths of trees in the outbreak areas and in the control is not described as typical dendrochronological. In particular, it makes no sense in the analysis to use such an important dendrochronological indicator as EPS (Expressed Population Signal), used to assess the synchronization of annual rings' width time series [31]. When the critical value of EPS ≥ 0.85, the chronology is considered sufficiently representative [32]. However, the EPS value does not describe the data presented in the present paper at all. Calculations of EPS for the studied sample plots show EPS values ranging from −0.55 (when the time series for damaged trees are in antiphase with time series of trees in the control stand) to 0.40. According to the principles of dendrochronological studies, these series cannot be considered synchronous and these data are not suitable for the construction of generalized dendrochronological indices. However, the direction of present research can rather be called, not dendrochronological analysis, but the analysis of dendroregulation processes. The rows of the annual rings' width first differences for trees in the outbreak zone and control group trees do not necessarily have to be synchronous. On the contrary, the proposed approach is based on the fact that these series are nonsynchronous, and this is how they differ.

As shown in Figure 4, the spectra of first differences series differ between the outbreak locations and controls. A comparison of the spectra between outbreak locations and control stands revealed that for most trees in the control, $f_{max}$ had the maximum possible value (Nyquist frequency $f$ = 0.5), while for the trees at the outbreak location, $f_{max}$ was much lower. Figure 5 shows the $s$ and $f_{max}$ values of the trees on sample plots pp04D and pp04K in the $\{s, f_{max}\}$ plane.

As illustrated in Figure 5, at small $s$ values, trees at the outbreak location differ from the controls by a smaller $f_{max}$.

Figure 6 shows the mean values of standard deviations of the first-difference series and mean frequencies of the spectrum maxima for sample plots pp04–pp11D and K.

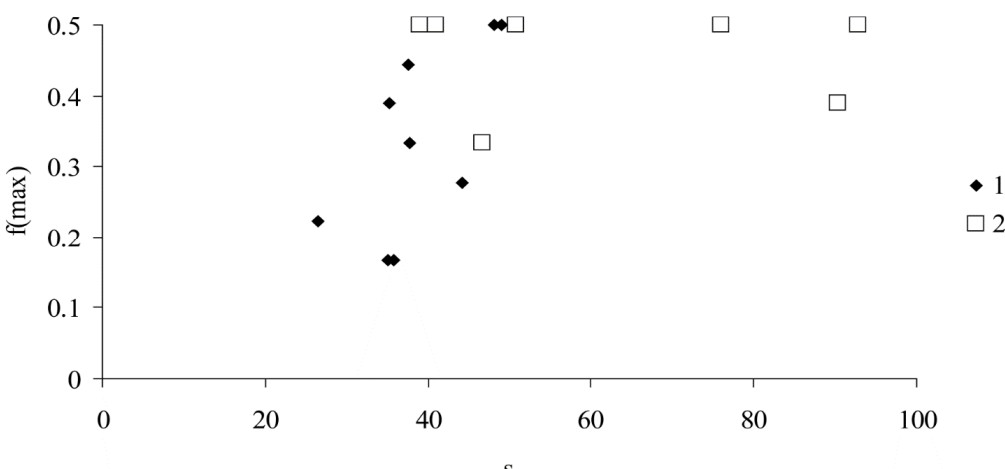

**Figure 5.** Characteristics of birch trees on plots pp04D (1) and pp04K (2) in forest stands of the Novosibirsk region.

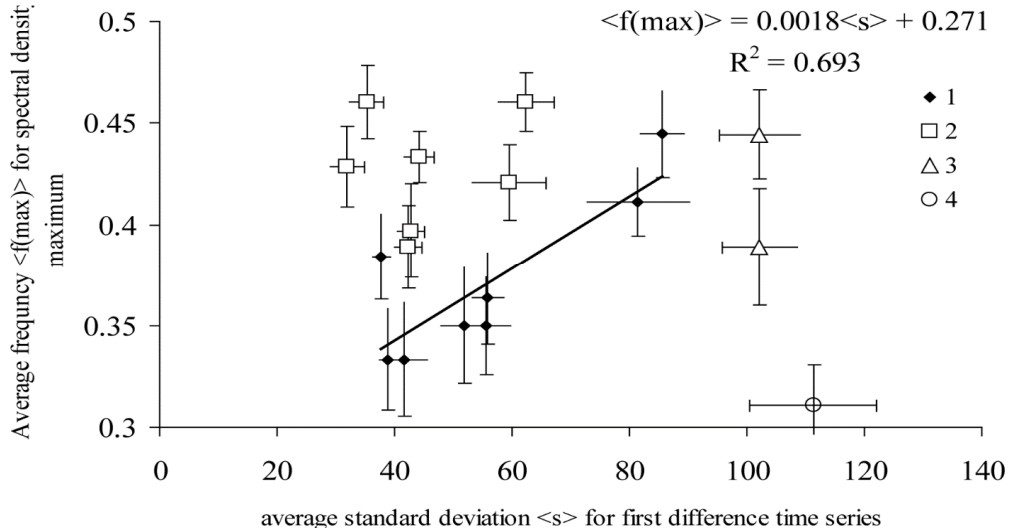

**Figure 6.** Average standard deviations of first-difference series and mean frequencies of spectrum maxima for sample plots pp04–pp11 D and K. 1: Outbreak locations, 2: control, 3: sample plots pp08K and pp09K, and 4: sample plot pp09D.

Figure 6 shows that if we exclude pp08K and pp09K (group 3), then for outbreak locations, the relation between $s$ and $f_{max}$ will be linear. With increasing $s$, $f_{max}$ will also go up, while in the control, no relation between $s$ and $f_{max}$ is discernable; for all control sample plots, at $f_{max}$ values between 0.4 and 0.5, $s$ values varied from 30 to 60. It can be theorized that the outlier plots pp08K and pp09K, whose characteristics deviated from the patterns observed for control plots, are located in areas with relatively severe damage, and these "control" plots should actually be regarded as small sites attacked by the spongy moth. As proof of this statement, we present a view of sample plot pp09K from an unmanned aerial vehicle, which clearly indicates damage to some tree crowns (the lower part of the photo in Figure 7).

Table 1 lists average standard deviations of series of first differences and mean frequencies of spectrum maxima for the sample plots at the studied locations of outbreaks of pest mass reproduction.

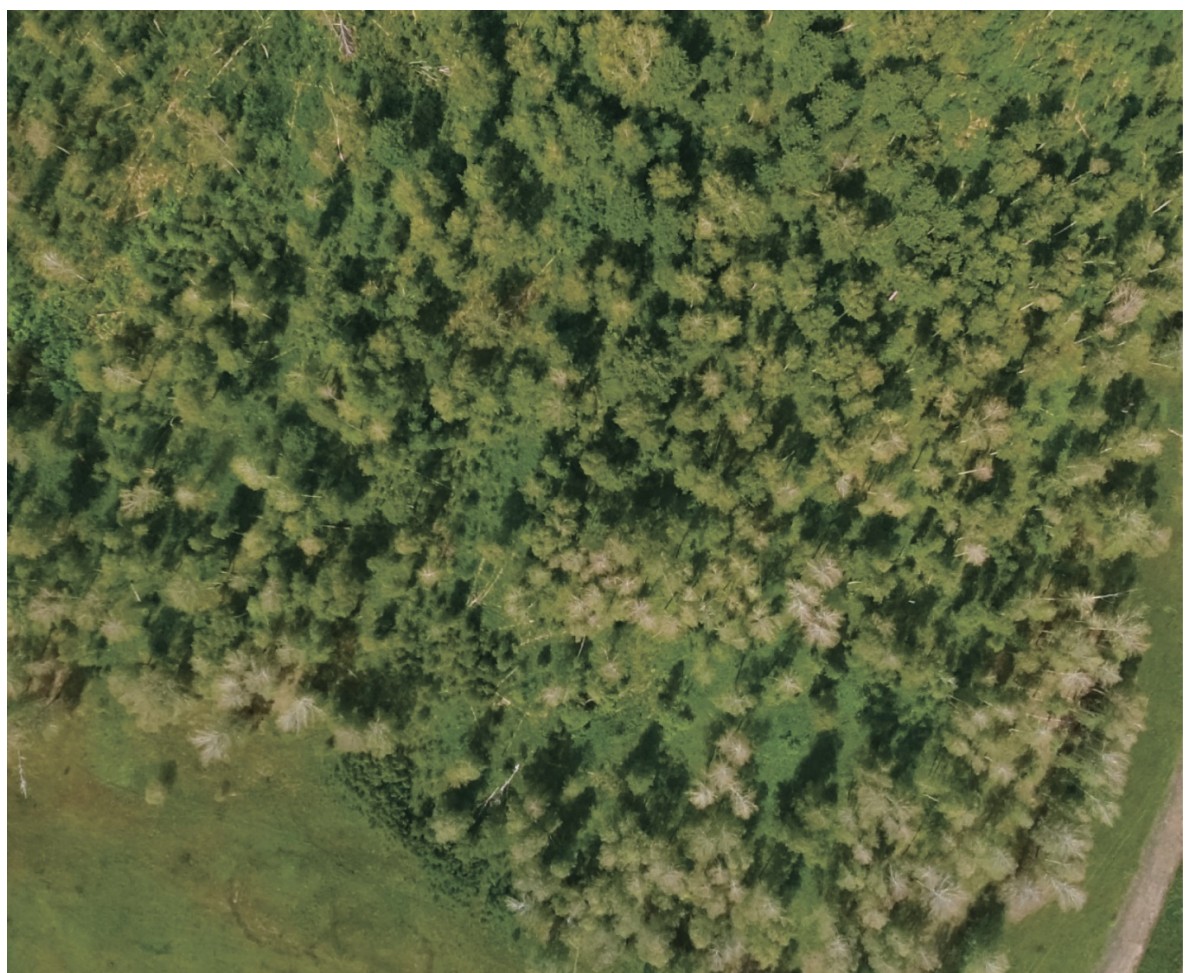

**Figure 7.** A photo of sample plot pp09K from an unmanned aerial vehicle.

**Table 1.** Average standard deviations ± standard error of first-difference series and mean frequencies of spectrum maxima ± standard error for sample plots at locations of outbreaks of pest mass reproduction (spongy moth, Siberian silk moth, and pine looper).

| Insect Species | Sample Plots | Outbreak Locations | | Control Stands | |
|---|---|---|---|---|---|
| | | StandardDeviation | Frequency | StandardDeviation | Frequency |
| Spongy moth | pp04D, pp04K | 38.79 ± 2.70 | 0.33 ± 0.05 | 62.39 ± 9.56 | 0.46 ± 0.028 |
| | pp05D, pp05K | 41.60 ± 8.14 | 0.33 ± 0.06 | 42.87 ± 4.68 | 0.40 ± 0.046 |
| | pp06D, pp06K | 55.92 ± 5.15 | 0.36 ± 0.45 | 42.25 ± 4.57 | 0.39 ± 0.041 |
| | pp07D, pp07K | 81.43 ± 17.39 | 0.41 ± 0.03 | 35.28 ± 5.77 | 0.46 ± 0.036 |
| | pp08D, pp08K | 85.47 ± 7.46 | 0.44 ± 0.04 | 102.14 ± 13.70 | 0.44 ± 0.044 |
| | pp09D, pp09K | 111.31 ± 21.77 | 0.31 ± 0.04 | 102.16 ± 12.88 | 0.39 ± 0.058 |
| | pp10D, pp10K | 55.59 ± 8.34 | 0.35 ± 0.05 | 59.61 ± 12.55 | 0.42 ± 0.037 |
| | pp11D, pp11K | 51.92 ± 7.87 | 0.35 ± 0.06 | 31.96 ± 5.66 | 0.43 ± 0.040 |
| Siberian silk moth | ip1D, ip1K | 44.50 ± 7.09 | 0.39 ± 0.035 | 64.54 ± 4.84 | 0.43 ± 0.033 |
| Pine looper | kp1D, kp1K<br>kp2D | 22.11 ± 1.67<br>29.08 ± 6.65 | 0.40 ± 0.03<br>0.39 ± 0.03 | 14.80 ± 1.62 | 0.42 ± 0.016 |

Similar calculations were performed for trees in the regions of Siberian silk moth and pine looper outbreaks (Figures 8 and 9).

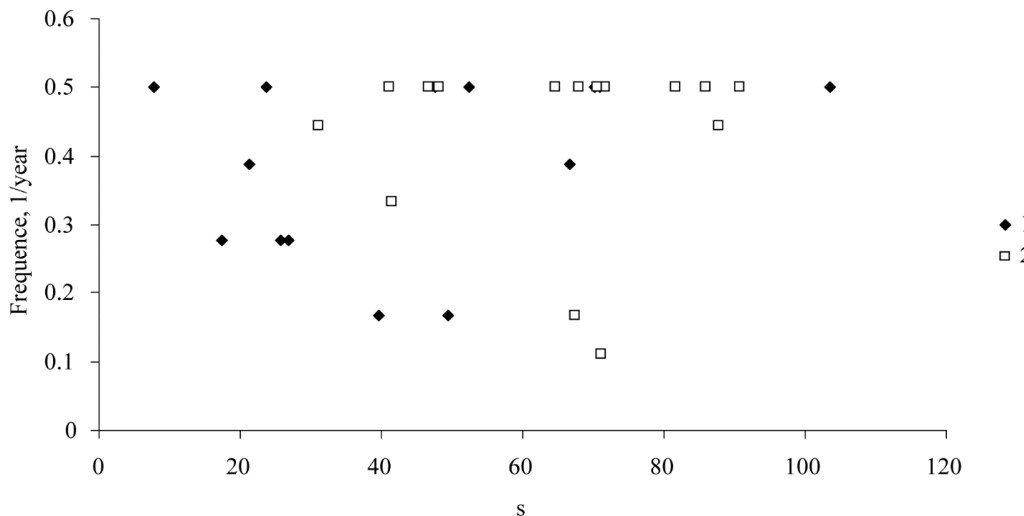

**Figure 8.** Parameters $s$ and $f_{max}$ for trees in a control intact forest stand (1) and at a location of a Siberian silk moth outbreak (2).

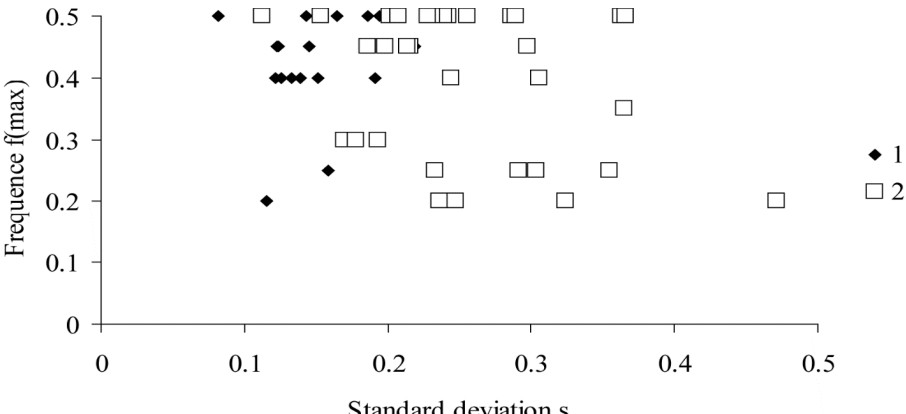

**Figure 9.** Parameters $s$ and $f_{max}$ of trees in the region of pine looper outbreaks. 1: Control, 2: outbreaks of the pest.

The quality of tree discrimination between an outbreak location and a control stand by means of $s$ and $f_{max}$ (Figure 9) was assessed using Wilks' $\lambda$ distribution. For the tree groups being analyzed, $\lambda$ was 0.602, with $F_{2,47} = 15.50$ and $p < 0.00001$. Table 2 is a classification matrix of linear discriminant analysis for trees in the outbreak region of the pine looper and in the control undamaged stand.

**Table 2.** The classification matrix for the *B. piniarius* outbreak region.

| Trees on Sample Plots | % of Correctly Classified Trees | Classified Trees | |
|---|---|---|---|
| | | Control | Damaged |
| Control | 76.5 | 13 | 4 |
| Damaged | 87.9 | 4 | 29 |
| Total | 84.0 | 17 | 33 |

Thus, from the results of the discriminant analysis, it can be concluded that with high significance, characteristics of the time series of first differences in TRW—at locations of the future pine looper outbreak for ~20 years before the outbreak—differed from such characteristics of trees in the control undamaged stand.

For each sample plot, $s$ and $f_{max}$ values normalized to the control stands were calculated next (Figure 10).

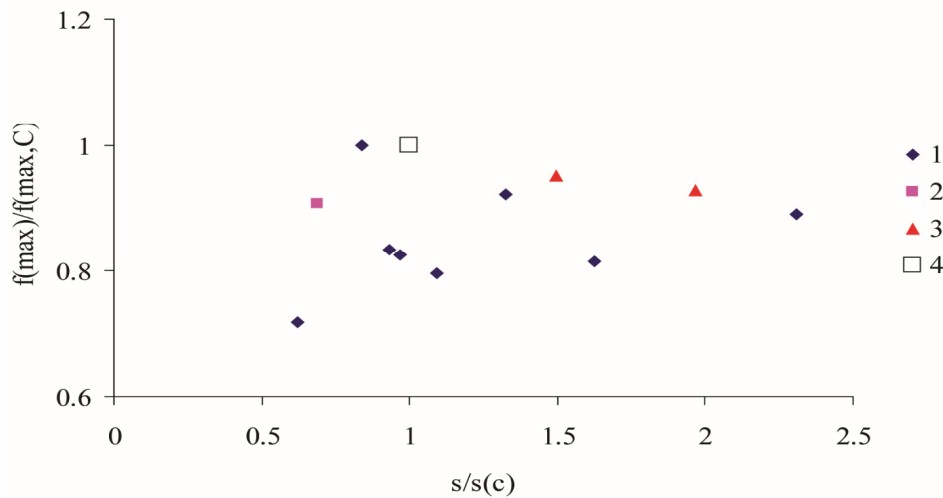

**Figure 10.** Values of $s_i$ and $f_{max,i}$ at outbreak locations, respectively, divided by $s_c$ and $f_{max,c}$ of control intact forest stands. 1: Spongy moth outbreak locations, 2: Siberian silk moth outbreak locations, 3: pine looper outbreak locations, and 4: control.

If $s_i$ and $f_{max,i}$ are the same between outbreak locations and control stands, then the normalized values are 1.0. As follows from Figure 10, some outbreak locations of the spongy moth and of the Siberian silk moth had slightly lower $f_{max}$ as compared to controls. For several spongy moth outbreak locations, $s$ turned out to be greater than that in the control, whereas $f_{max}$ was slightly lower relative to the control. The greater $s$ values than in the control for pine looper outbreak locations may be due to the fact that the core samples from trees at these outbreak locations were taken decades after the outbreak period (in 2015, whereas the outbreak happened in the mid-1970s), i.e., when trees heavily damaged at outbreak locations had already died and it became impossible to collect core samples from them. Only trees with minor damage survived, in which the crown recovered after the damage.

## 4. Discussion

How can one explain the differences in regulatory characteristics of first-difference series of TRWs between pest outbreak locations and undamaged forest stands? For an explanation, it is necessary to take advantage of the well-known (in statistical physics) concept of fluctuation–dissipation relations within systems. How can an investigator interpret the FDT in terms of ecological processes in a forest? The series of first differences in TRWs can be considered characteristics of regulatory processes in trees during their growth. Any regulatory processes can be characterized with the help of two parameters: (i) characteristic time of regulation, which is inversely proportional to the frequency of the maximum in a spectrum of the system's parameters, and (ii) deviations from a set value of a characterizing variable, which are calculated as standard deviation of a series of first differences.

Regulatory processes in series of first differences in widths of an increment of an annual ring can be regarded as some high-frequency fluctuations of relatively slow (taking decades) tree growth processes. The higher the frequency at the peak of a spectral function, and the lower the standard deviation of regulated series (such parameters are typical for trees in the control sample plots), the faster the system will recover from deviations from the norm. From Equation (1), it follows that FDT guarantees a connection between (a) characteristics of the spectrum before an external factor acts on the system and (b) the response of system after this action. Thus, it can be hypothesized that in control undamaged forest stands trees feature a rapid but not very strong response to possible insect attacks, as well as the development of antibiosis reactions.

One can propose a simple classification of trees by the type of spectral characteristics and categorize all trees into four groups: tree group 1 with small $f_{max}$ and $s$; tree group 2 with small $s$ and large $f_{max}$; tree group 3 with large $s$ and $f_{max}$; and finally, tree group 4 with large $s$ and small $f_{max}$.

Trees from groups 3 and 4 reacted to external factors slowly, whereas trees from groups 1 and 2 reacted quickly, but the amplitude of the reaction was greater in tree group 2 than in tree group 1. The stablest type of tree was in group 1, followed by trees from groups 2 and 3, and lastly group 4.

If there is a relationship between the susceptibility of a tree to insects and regulatory characteristics of phytomass growth, then a question arises: How do insects find out which trees grow in a given stand? Trees quickly responding to external factors (and unsuitable as food for the development of caterpillars) or trees with a delayed reaction to the leaf damage by insects? Obviously, insects are not able to directly measure TRWs and evaluate the spectra of series of the widths' first differences. A possible answer to this question is the existence of a connection between the regulatory processes of tree growth and a release of various chemicals by a tree. For instance, drooping birch is known to be able to release volatile monoterpenes in response to severe damage by silkworm caterpillars [33]. It must be noted that even minor types of damage induce a birch response, which can subsequently affect insect physiology [34]. These signals can be read by female moths for oviposition in the respective forest stands. Nonetheless, in the case of the spongy moth, this statement is valid only for the biotype featuring flying females [35]. In the case of flightless females, the main contribution to dispersal-related activities of the biotype is made by the ballooning of caterpillars [36]. Probably, for this biotype, the choice of plants is determined by the caterpillars' testing of leaves in the spring, and their subsequent fixing of themselves on the selected trees or continued ballooning until a more suitable forest stand is chosen. The emission of volatile organic compounds during damage by insects is also observed in coniferous trees, both in pine [37] and in spruce species [38]. Of course, these changes may be a consequence of insect actions instead of preceding an outbreak. Unfortunately, it is not possible to determine prior to an outbreak which trees will be susceptible and which will not, and to carry out appropriate chemical assays in advance. Nevertheless, the mechanism in question may work in the forests analyzed in our paper. Furthermore, the variability of forest stands in terms of susceptibility can also determine the ability of an insect population to gain a foothold in a given stand, and may allow pests to realize their potential maximally in the case of "melancholic" and "phlegmatic" trees, or only to a limited extent in "choleric" or "sanguine" trees. If there are differences in concentrations or ratios of terpenes between future outbreak locations and attack-resistant stands, then these parameters can probably be detected by adult insects looking for a place to lay future eggs. Consequently, insects can possibly evaluate the quality of food and ultimately the fate of their own offspring by choosing trees and stands by means of regulatory parameters of a tree's response to leaf loss. To select an individual tree, insects must have sensory systems with a high spatial resolution, whereas navigation using summarized parameters of trees in a forest stand requires less accuracy (this supposition explains the presence of a few quickly responding trees having $f_{max}$ close to $f = 0.5$ at the outbreak locations).

## 5. Conclusions

Thus, with the help of a retrospective analysis, we demonstrated that forest stands susceptible to damage by defoliators and non-susceptible stands differ in characteristics of the growth of annual rings. The studied TRWs data series ended 2 years before the future population outbreak. Thus, the obtained differences between the damaged and control trees are a predictor of future damage to the stand by pests for 2–4 years, when there were no external manifestations of tree weakening yet. The approach proposed here makes it possible to explain the existence of small sites of forest insects' population outbreaks. Given that insects cannot measure the characteristics of annual rings in trees but are able to react to the chemical compounds emitted by trees (for example, terpenes), the relationship

identified between characteristics of first-difference series of TRWs and an opportunity for insects to implement a population outbreak in certain forest stands indicates the following. There are possible specific connections between the characteristics of growth-regulatory processes in trees and the profile and concentrations of volatile molecules (attractants or insect repellents) released by a tree and/or nonvolatile compounds that determine plants' susceptibility to phytophages. Evidently, the uncovered relationships can be utilized both for monitoring the state and resistance of trees to attacks by forest insects, and for finding attractants and repellents for certain species of forest insects.

**Author Contributions:** Conceptualization, V.S. and V.M.; Methodology, V.S., P.K., A.K., I.S., O.T., Y.I. and Y.A.; Software, A.K.; Validation, A.K. and Y.I.; Formal analysis, P.K., A.K., I.S. and O.T.; Investigation, P.K., A.K., I.S., O.T., Y.I., Y.A. and V.M.; Resources, Y.A. and V.M.; Data curation, V.S.; Writing—original draft, V.S.; Writing—review & editing, V.M.; Supervision, V.S.; Project administration, V.M.; Funding acquisition, V.M. All authors have read and agreed to the published version of the manuscript.

**Funding:** This research was supported by the Russian Science Foundation (grant #21-46-07005 to V.M. for the *L. dispar* part of the research and grant #22-24-00148 to V.S. for parts of the study dealing with *D. sibiricus* and *B. piniarius*).

**Conflicts of Interest:** The authors declare no conflict of interest.

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
