# Peer review of "Differentiation of Forest Stands by Susceptibility to Folivores: A Retrospective Analysis of Time Series of Annual Tree Rings with Application of the Fluctuation-Dissipation Theorem"

_forests, doi:10.3390/f14071385_

Round 1

Reviewer 1 Report

The manuscript introduced the fluctuation-dissipation (physical theory) into the insect detection. The authors collected a number of cores with and without insect influences and the innovation of this work is also highlighted. Due to the following reasons, I determine to give the manuscripta major revision.

1.     In the introduction part, the authors realized there were age trends among the tree rings. The authors applied the first differences to avoid the tendency. In general, the individual tree-ring measurement series were detrended with ARSTAN (Speer, 2010). I hope the authors can cite some papers with first difference approach or may explain why the research did not use the curves to remove the trend.

2.     The manuscript directly used TRW or the differences of TRW to detect the defoliators. And readers also want to know whether the study period (2000 - 2020) has good expressed population signal (EPS) values. In the updated version, I hope the authors might provide the EPS values for the three insect detections.

3.     The manuscript list four types of tree species: sanguine, choleric, phlegmatic and melancholic. In the discussion part, I hope the authors can explain more about whether the classification of these four types refer to the specific defoliator or the general defoliators.  

Speer, J. H. (2010). Fundamentals of tree-ring research. University of Arizona Press.

Author Response

Dear Reviewers,

Thank you for taking time to review the manuscript. According to your suggestions, we had fully revised the manuscript. Because the manuscript has been considerably revised, most of changes have not been visualized for readability. The reviewers' comments did not include point lines; we have presented summary responses to the comments.

  1. We have added a description of the differences of our method from the standard methods of dendrochronology. In particular, it is shown that EPS criteria is not informative for this type of research (Result section). A description of obtaining a stationary series for further spectral analysis by the first difference method has also been added (Mat&method section).
  2. The "literary" names of four classes of trees depending on the calculated parameters were removed from the discussion. Perhaps this will simplify the perception of the material.
  3. Habitat descriptions of study species and stand's characteristics have been added (Mat&method section).
  4. Differences in the scope of the experiment for different species were determined by the difficulty of conducting work for forest species in the conditions of taiga forests. This is added to the text of MS (in the beginning of Mat&method section).
  5. All major species of both forest stands and insect pests of Eastern Eurasia are listed in the work. For example, Quercus spp. is not common and is not damaged in this region.
  6. Unified design of figures, text style and design of references.

Thank you again for your suggestions! Hopefully the revision could make the manuscript be improved and be easily understood.

Regards, Authors.

Reviewer 2 Report

The paper presents interesting results in time series dendrochronological analyses of forest stands in the region of southern Siberia.

- Add a chapter describing natural conditions - describe the broader spatial relationships (is this the southern edge of the boreal forest biome?), with the occurrence of which stands? 

- I recommend writing Betula pendula instead of the English name "birch" for better readability and understanding. For example, "cedar" here is Pinus sibirica.

- In the Introduction, briefly describe the bionomics, distribution of Lymantria dispar and other species, what is their basic strategy here in the boreal biome

- include a separate chapter describing the natural conditions, detailed description of the sites in the form of a table. Any data on the habitat - soils, trophic and water conditions of the stands are missing. Were these the same conditions?

- fir-cedar-birch - better to replace Abies sibirica-Pinus sibirica-Betula

- in the text of the results of the site e.g. pp03D list in words what site it is. Unintelligible for the reader

- better would be to evaluate the same number of sites for a given insect species, unless it is useful to emphasize that Dendrolimus pini is based on only one pair of sites, etc.

- to include in the discussion a comparison with other areas where the insect pest species in question occur, or are the results applicable to the whole Euro-Siberian area? Will other woody species respond in the same way? For example, species of Quercus spp.

- Graph descriptions to be unified

- Unify styles in Literature

Author Response

(The authors gave the same response as above.)

Round 2

Reviewer 1 Report

The author answered all the questions that I have. I hope that the authors can improve some tiny language problems.